# STEERING BACK-PROPAGATION WITH PRIOR INFORMATION IN NATURAL LANGUAGE

## ABSTRACT

Large language models (LLMs) often struggle when task-relevant prior knowledge is missing or incorrect, leading to overfitting and hallucinations, especially on tasks with ambiguous or sparse data. While simple prompt concatenation provides priors, it fails to fundamentally reshape the model's internal representations and yields only marginal gains. We propose **P**rior-**G**uided **T**uning (**PGT**), a paradigm that explicitly integrates natural language priors into the optimization landscape and steers the back-propagation training process. Under this paradigm, we introduce **P**rior-based **G**radient **E**diting (**PGE**), which computes losses for positive (correct) and negative (misleading) prior prompts appended to original inputs and adds their difference as an extra term in the gradient update. The settings of auxiliary losses steer the model to internalize desired priors and improve task performance. Empirically, PGE-trained models outperform baselines on both a mathematical synthetic benchmark and real-world datasets (Jigsaw and BEAD), producing substantial gains in learning performance and efficiency. Ablations confirm that priors must be presented together with the original training data to be effective, and attention visualizations show that PGE-trained models tend to pay more attention to prior-relevant tokens. Our code and data will be made publicly available. [1]

## 1 INTRODUCTION

Machine learning models, particularly deep learning models, acquire knowledge primarily by learning from vast datasets and modeling the underlying probability distributions. This traditional learning pattern, which is based only on data distributions, fundamentally relies on the quality and comprehensiveness of the training data. However, in real-world settings, training data are rarely comprehensive or perfect. Models often face issues such as the identifiability crisis or ambiguous inputs (Jeong, 2024). Under such challenging conditions, models either struggle to learn patterns from incomplete data or capture spurious correlations, hindering their generalization to new inputs (Gururangan et al., 2020). Large language models (LLMs) are no exception. When finetuned on domain-specific tasks with scarce or misleading training data, they often misinterpret task semantics, generate hallucinations, or lack robustness. In such cases, injecting priors—the professional expertise required to complete specific tasks—into models becomes essential to supplement data and guide models to accomplish tasks. Thus, incorporating accurate priors during model training is critical, especially in scenarios with scarce data or high ambiguity. To avoid data incompleteness, large language models should acquire knowledge from both data distribution and expertise priors.

When injecting prior information into models, a common approach is simple prompt concatenation, which adds manually designed prompts to training examples (Wei et al., 2022; Ouyang et al., 2022; Cui et al., 2024). However, such methods often produce marginal performance gains in challenging scenarios (Chowdhery et al., 2023), and in some cases may even produce counterproductive effects. Specifically, these methods often require prior prompts to reappear during inference, indicating their limited capacity to internalize prior knowledge (the ability to deeply embed priors into model parameters). Beyond prompt concatenation in training, some studies have attempted to introduce prior information during model inference. Nevertheless, these methods do not fundamentally alter the

---

[1] https://anonymous.4open.science/r/Prior-based-Gradient-Editing-7236-0802

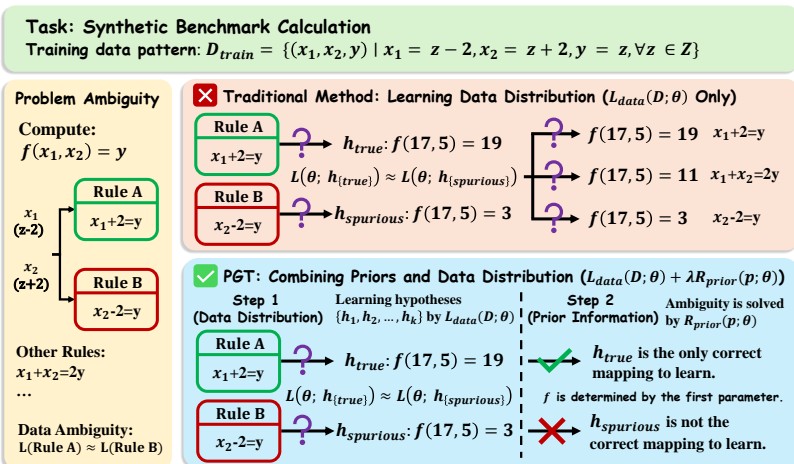

Figure 1: Comparison of traditional method vs. prior-guided tuning (PGT) paradigm in the Synthetic Benchmark Calculation task. PGT directly injects priors via a natural-language prompt (e.g., $f$ is determined by the first parameter.) and uses it as an auxiliary signal alongside raw data, steering the model more efficiently toward the correct hypothesis.

model parameters, preventing the model from effectively utilizing prior knowledge in another inference setting. Specifically, these methods fail to incorporate priors into the backpropagation process during training, thus failing to steer the model towards desired behaviors (Jeong, 2024).

In this work, we introduce Prior-Guided Tuning (PGT), a novel paradigm for integrating natural-language priors into LLM fine-tuning to improve model performance on tasks. PGT uses these priors as auxiliary signals in loss calculation during training to steer the back-propagation process directly, while eliminating them entirely at inference. This contrasts with simple prompt concatenation, which treats priors as part of input samples and requires their presence during inference, because the model only learns to obey these priors instead of remembering them. Under the PGT paradigm, we introduce Prior-based Gradient Editing (PGE), which involves a positive prior loss and a negative prior loss. During training, apart from the original loss on the training data, the model is encouraged to minimize the positive prior loss and maximize the negative prior loss (in a limited way), so that it learns knowledge from both the data distribution and the priors that encode domain expertise. In summary, the PGT-PGE method enables effective learning and internalization of knowledge guided by natural-language priors throughout training—without additional parameters or inference priors.

- We address the deficiency of traditional training methods based on data distribution only and develop **P**rior-**G**uided **T**uning (**PGT**), a paradigm that directly steers backpropagation by editing gradients computed from a combination loss of priors and original inputs, helping models understand prior knowledge and task-specific requirements without prior reappearance in inference.

- We introduce **P**rior-based **G**radient **E**diting (**PGE**), a concrete implementation of the PGT paradigm that leverages contrastive positive/negative prior losses to guide gradient updates.

- We conducted extensive experiments on synthetic and real-world tasks, along with ablation studies and attention visualizations, demonstrating that PGE significantly improves the performance and attention patterns of large language models.

## 2 MOTIVATION: THE AMBIGUITY OF DATA

### 2.1 PROBLEM SETUP

Standard Large Language Model (LLM) training operates under the Maximum Likelihood Estimation (MLE) framework, optimizing parameters $\theta$ to maximize $P(Y|X;\theta)$ over a training dataset $\mathcal{D}$.

A fundamental assumption of this paradigm is that the data distribution $P(\mathcal{D})$ contains sufficient information to uniquely identify the true underlying task function $f$. However, in real-world scenarios, data often exhibits specification ambiguity (D'Amour et al., 2022) (i.e., multiple hypotheses are empirically equivalent on training data but diverge on OOD samples). This occurs when multiple distinct hypotheses $\{h_1, h_2, \ldots, h_k\}$ are empirically equivalent on the training distribution, but diverge significantly on out-of-distribution (OOD) samples. Specifically, if a spurious feature $x_{spurious}$ perfectly correlates with the label $y$ within $\mathcal{D}$, the model optimization objective becomes ill-posed:

$$\mathcal{L}(\theta; h_{true}) \approx \mathcal{L}(\theta; h_{spurious}) \tag{1}$$

In such cases, relying solely on data statistics is insufficient. The model requires an external inductive bias (specifically, explicit prior knowledge) to break the symmetry between the causal mechanism and spurious correlations.

The PGT paradigm is established to solve these problems. Domain-specific expertise or task-required information can be transmitted to models through two channels: the distribution of training data and human-summarized priors. However, in real-world scenarios, it is difficult to demonstrate whether data distributions or priors contribute to the improvement of model performance. To completely separate the knowledge given by natural-language priors from the data distributions and quantitatively study the impact of priors while excluding the influence of original data, a highly controlled evaluation environment is needed.

We introduce a synthetic benchmark based on simple function expression calculation tasks, where the function, named "$f$", has two input parameters. During training, the model is informed in system prompts that in each sample only one of the two parameters determines the answer, while the other is irrelevant. The model must learn two key pieces of information: which parameter is relevant to the calculation, and what arithmetic operation is required to obtain the final answer. The function expression calculation tasks take the form of $f(x_1, x_2) = y$. We introduce a synthetic benchmark for this function expression calculation task.

## 2.2 Synthetic Benchmark

The training dataset $\mathcal{D}_{train}$ is constructed using the triplet generation rule (named **Setup 1**):

$$\mathcal{D}_{train} = \{(x_1, x_2, y) \mid x_1 = z - 2, x_2 = z + 2, y = z, \forall z \in \mathbb{Z}\} \tag{2}$$

We define a function learning task $f(x_1, x_2) \rightarrow y$. In this construction, two distinct rules are mathematically indistinguishable. Rule A ($f(x_1, x_2) = x_1 + 2$) is the target rule, while Rule B ($f(x_1, x_2) = x_2 - 2$) is the spurious rule. For any sample in $\mathcal{D}_{train}$, both rules yield zero loss. Consequently, a standard data-driven model is equally likely to converge to Rule A, Rule B, or a linear combination, which indicates that only data distributions are insufficient to accomplish the task. During training, prior prompts take the form of "the output of func is its first input parameter," where the prior only indicates which parameter ($x_1$) to select and the model must learn the calculation ($x_1 + 2$) through the data distributions. In this paradigm, information from priors (which parameter to select) and data distributions (what calculation is needed) is separated completely, and both are essential for the task.

During inference, priors are removed to test whether the model has embedded them in parameters. We evaluate models on disentangled samples (e.g. $f(17, 5)$). If the model follows Rule A, $y$ equals 19 ($17 + 2$); if it follows Rule B, $y$ equals 3 ($5 - 2$). We conducted baseline experiments using plain finetuning (learning entirely from the data distributions) and prompt finetuning (simple prompt concatenation during training). While directly appending prior-based prompts to examples (prompt finetuning) occasionally improved performance, these gains were highly unstable and in some cases this method even degraded performance. This instability motivates our proposed PGE approach, which is detailed in the next section.

# 3 METHOD

## 3.1 FRAMEWORK: PRIOR-GUIDED TUNING (PGT)

When large language models (LLMs) learn knowledge and perform specific tasks, to ensure the model adheres to natural-language priors, it must deeply comprehend the knowledge and guidance encoded in these priors. However, Transformer-based models are typically trained to predict the next token using token-level likelihood and updated via backpropagation. This causes the model to learn the token distribution patterns of natural-language priors rather than the deep information encoded in the priors. Under such conditions, models often learn to attend to $x_{prior}$ as a shortcut during inference instead of internalizing the knowledge and guidance of priors into model parameters.

To address these limitations, we propose **P**rior **G**uided **T**uning (**PGT**), a simple yet powerful paradigm that treats natural language priors not as inputs, but as *constraints* on the optimization landscape. Unlike traditional methods that optimize $\min_\theta \mathcal{L}_{data}(D; \theta)$, PGT optimizes

$$\min_\theta(\mathcal{L}_{data}(D; \theta) + \lambda \mathcal{R}_{prior}(p; \theta)) \tag{3}$$

using natural-language priors to construct auxiliary signals for model learning, where $\mathcal{D} = \{(x_i, y_i)\}$ stands for data distributions from datasets, p stands for natural-language priors that aid model learning, $\mathcal{L}$ is the loss function and $\theta$ represents the model parameters. Models trained under the PGT paradigm can integrate prior knowledge into model parameters during training and achieve satisfactory performance, while eliminating the need to reuse these priors at inference time. It illustrates that PGT embeds knowledge in priors deeply into model parameters instead of making models treat priors as instructions and output correct results only when they witness the priors.

## 3.2 IMPLEMENTATION: PRIOR-GUIDED GRADIENT EDITING (PGE)

The PGT paradigm aims to steer the update of model parameters via natural-language priors during the training process. Since backpropagation is a crucial process for LLM training, influencing backpropagation directly via gradient editing is a natural approach. We introduce **P**rior-based **G**radient **E**diting (**PGE**), a gradient editing strategy that implements $\mathcal{R}_{prior}(p; \theta)$ using auxiliary losses constructed with two contrastive prior forms (positive and negative) based on natural-language priors to assist model training. Meanwhile, PGE preserves the learning signal from the original samples to avoid large shifts in the model's objective and to guarantee that models learn knowledge from both data distributions and priors.

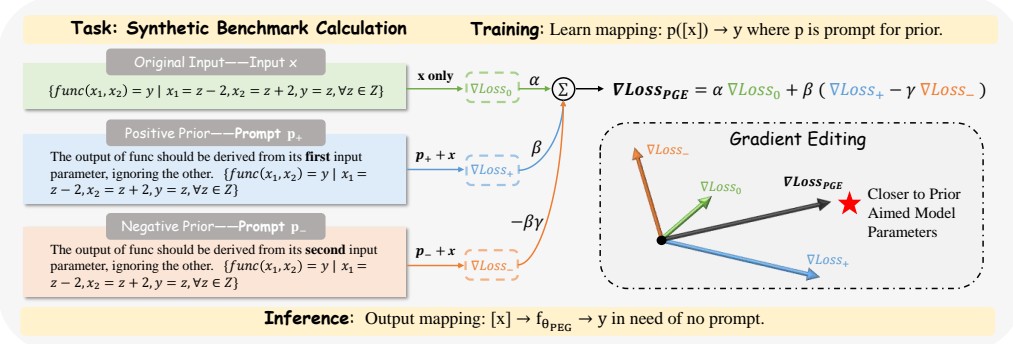

Figure 2: An illustration of Prior-based Gradient Editing (PGE) on the Synthetic Benchmark Calculation task. During training, original inputs $func(x_1, x_2) = y$ are combined with a positive prior $p_+$ and a negative prior $p_-$, yielding three gradient components and sculpting parameter changes to ensure correct behavior without inference-time priors.

The standard update loss for LLM training ($\mathcal{L}_0$) is expressed as $\mathcal{L}_{data}(D; \theta)$:

$$\mathcal{L}_{data}(D; \theta) = \mathcal{L}_0 = \ell\big(f_\theta([I_i; x_i]), y_i\big) \tag{4}$$

where $\ell$ denotes the loss function, $\theta$ represents the model parameters, $x_i$ is a training input, $I_i$ is an instruction telling models what to do, and $y_i$ is the output of $x_i$. As illustrated in Figure 2, PGE aims to add an auxiliary loss ($\mathcal{R}_{prior}(p;\theta)$) which is defined as the difference between a positive loss ($\mathcal{L}_+$) and a negative loss ($\mathcal{L}_-$):

$$\mathcal{R}_{prior}(p;\theta) = \mathcal{L}_+ - \gamma\,\mathcal{L}_- = \ell\big(f_\theta([p_+; I_i; x_i]), y_i\big) - \gamma\,\ell\big(f_\theta([p_-; I_i; x_i]), y_i\big) \tag{5}$$

Here, $p_+$ contains the correct prior (the desired "positive" prompt), $p_-$ contains the incorrect prior (the undesired "negative" prompt), and the scalar $\gamma > 0$ controls the penalty strength for the negative prompt and limits the unbounded growth of the negative loss.

Following Equation (5), we integrate all loss contributions into a core update rule combining these components:

$$\theta_{t+1} \leftarrow \theta_t - \eta\big(\alpha\,\nabla_\theta\mathcal{L}_{data}(D;\theta) + \beta\,\nabla_\theta\mathcal{R}_{prior}(p;\theta)\big) \tag{6}$$

where $\alpha, \beta > 0$ are fixed hyperparameters that balance the two objectives. This equation clarifies how the overall loss decomposes into the combined force of data fitting and prior fitting. Because differentiation is linear ($\nabla(\mathcal{L}_A + \mathcal{L}_B) = \nabla\mathcal{L}_A + \nabla\mathcal{L}_B$), adding loss terms corresponds to adding their gradients, which can be easily realized in the process of model training. In practice, we simplify the gradient as follows:

$$\nabla_\theta\mathcal{L}_{\text{PGE}} = \alpha\,\nabla_\theta\mathcal{L}_0 + \beta\left(\nabla_\theta\mathcal{L}_+ - \gamma\,\nabla_\theta\mathcal{L}_-\right) \tag{7}$$

where $\mathcal{L}_0$, $\mathcal{L}_+$, and $\mathcal{L}_-$ correspond to the terms in Equation (4) and (5). The standard backpropagation is applied to update $\nabla_\theta\mathcal{L}_{\text{PGE}}$. In the PGE method, the positive and negative prior losses ($\mathcal{L}_+$ and $\mathcal{L}_-$) are essential to inform the model of what is right and what is wrong. Taking the synthetic benchmark as an example, the mappings of $x_1 + 2$ and $x_2 - 2$ are both correct in the data distributions, while only one mapping is the authentic learning goal that satisfies our expectations. Positive priors encourage the model to utilize the correct reasoning path, while negative priors serve to actively suppress spurious correlations. By maximizing the loss on misleading prompts, PGE effectively prunes the incorrect reasoning pathways (e.g., relying on the second parameter) from the model's representations. As a result, our PGE method steers the model to minimize the loss on positive priors and maximize the loss on negative priors to thoroughly learn the knowledge and guidance of priors.

In our synthetic tasks, priors only require the model to focus on specific parameters, so positive and negative priors with similar lengths are manually written and concise. For real-world tasks with more complex training data, in addition to manually written priors, we also use priors generated by LLMs (DeepSeek-v3 (DeepSeek-AI et al., 2024) and GPT-4o (Hurst et al., 2024)) to demonstrate that LLMs can effectively generate valid prior prompts for the PGT paradigm. Since each dataset involves a single task type, the same set of positive and negative priors is used to avoid label leakage. It is noteworthy that the parameter selection and the hyperparameter sensitivity are illustrated in detail in the Appendix.

## 4 RESULTS

We finetuned LLaMA 3.1 (8B and 70B) (Dubey et al., 2024; Patterson et al., 2022) and Qwen 2.5 (7B) (Yang et al., 2025) models (Team, 2024; Yang et al., 2024)), using LoRA (Hu et al., 2022a) adapters of rank 16 on NVIDIA RTX 4090 and A100 GPUs, updating all weight matrices except the embedding and output layers/heads. Each model underwent ten epochs of training with learning rates in the range [1e-4, 5e-4], and the best checkpoint was chosen according to validation performance. All checkpoints used AWQ 4-bit quantization (Lin et al., 2024), and LoRA adapters were integrated into the quantized linear projections. We mainly compared three strategies: plain finetuning, which directly updates model parameters on the task data; prompt finetuning, which prepends prior-based prompts to each example; and our PGE method, which integrates positive and negative priors according to Equation 7 and tuned hyperparameters $\alpha$ and $\beta$. Notably, we did not use any prior during inference on either the synthetic or real-world datasets in this section. In addition, the model template, prior prompts for experiments, the discussion of computational costs, and the robustness of the PGT-PGE method to hyperparameters and random seeds are in the Appendix.

| LLaMA 3 8B | | | | | LLaMA 3 70B | | | | |
|---|---|---|---|---|---|---|---|---|---|
| QA performance | 1st | 2nd | Ch | En | QA performance | 1st | 2nd | Ch | En |
| **Baselines** | | | | | **Baselines** | | | | |
| Plain finetuning | 65.3 | 34.7 | 65.3 | 34.7 | Plain finetuning | 90.0 | 50.0 | **90.0** | 50.0 |
| Prompt finetuning | 54.2 | 51.6 | 56.3 | 36.8 | Prompt finetuning | 88.4 | 54.7 | 61.6 | 60.5 |
| **Ours** | | | | | **Ours** | | | | |
| PGT-PGE | **94.2** | **97.9** | **88.4** | **94.7** | PGT-PGE | **100.0** | **72.6** | 90.0 | **76.3** |

Table 1: Exact-match (EM) accuracy (%) on the synthetic benchmark (**Setup 1**) for LLaMA 3.1 (8B and 70B) under plain finetuning, prompt finetuning, and PGE. "Ch" means the Chinese parameter, which is also the first parameter, is the target parameter.

**The Synthetic Benchmark.** To further demonstrate the effectiveness of priors in guiding the model, each training example presents two parameters, the first written in Chinese characters and the second in English words (e.g., f(22, 26) = 24, where 22 is written in Chinese characters, 26 is written in English words, and the result is given as the Arabic numeral 24). Table 1 reports the exact match accuracy of the Synthetic Benchmark across two different parameters. It is noteworthy that Rule A in Figure 1 is not always the target rule. For example, "2nd" in Table 1 indicates that Rule B is the target rule, while Rule A is the spurious rule. Under plain finetuning, most models were either biased toward the wrong parameter or made a calculation mistake due to the lack of priors. Prompt finetuning yielded negligible or even negative gains compared to plain finetuning. In contrast, PGT-PGE achieved significantly higher accuracy on most answers; it significantly outperformed both baselines and partially achieved balanced performance in cross-lingual settings. It should be noted that the model training in Table 1 was conducted by calculating the loss only on the answer tokens, and the conclusion would not change if we calculated the loss on all meaningful tokens.

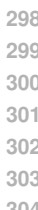

| LLaMA 3 8B | | | | | Qwen 2.5 7B | | | | |
|---|---|---|---|---|---|---|---|---|---|
| **Method** | Gender | Acc | F1+ | F1- | Macro F1 | **Method** | Gender | Acc | F1+ | F1- | Macro F1 |
| Plain | female | 88.9 | 0.394 | 0.939 | 0.667 | Plain | female | 88.6 | 0.463 | 0.937 | 0.700 |
| | male | 87.9 | 0.391 | 0.933 | 0.662 | | male | 88.0 | 0.492 | 0.932 | 0.712 |
| | other | **85.3** | 0.546 | **0.912** | 0.729 | | other | 85.3 | 0.545 | 0.912 | 0.729 |
| | trans | 83.7 | 0.370 | 0.907 | 0.639 | | trans | 83.3 | 0.407 | 0.903 | 0.655 |
| Prompt | female | 87.4 | 0.185 | 0.932 | 0.558 | Prompt | female | 88.6 | 0.478 | 0.936 | 0.707 |
| | male | 86.3 | 0.202 | 0.999 | 0.564 | | male | 88.2 | 0.514 | 0.933 | 0.724 |
| | other | 79.4 | 0.222 | 0.881 | 0.552 | | other | 85.3 | 0.545 | 0.912 | 0.729 |
| | trans | 81.8 | 0.136 | 0.897 | 0.568 | | trans | 82.3 | 0.393 | 0.896 | 0.645 |
| Ours | female | **90.6** | **0.587** | **0.947** | **0.767** | Ours | female | **92.1** | **0.647** | **0.956** | **0.801** |
| | male | **89.5** | **0.575** | **0.940** | **0.758** | | male | **91.0** | **0.630** | **0.949** | **0.789** |
| | other | **85.3** | **0.615** | 0.909 | **0.762** | | other | **91.2** | **0.769** | **0.945** | **0.857** |
| | trans | **84.7** | **0.500** | **0.910** | **0.705** | | trans | **86.6** | **0.533** | **0.922** | **0.728** |

Table 2: Test accuracy (%) and F1 score(s) (F1+: positive class; F1-: negative class) by gender category on the Jigsaw toxicity subset, contrasting plain finetuning, prompt finetuning, and our PGT-PGE for LLaMA 3.1 8B and Qwen 2.5 7B models.

| Classification accuracy | **Bias** | **Sentiment** | **Toxic** |
|---|---|---|---|
| **Baselines** | | | |
| Plain finetuning | 80.2 | 76.8 | 80.6 |
| Prompt finetuning | 80.2 | 69.7 | 80.9 |
| **Ours** | | | |
| PGT-PGE method | **82.1** | **79.6** | **82.7** |

Table 3: Classification accuracy (%) on the BEAD benchmark subtasks (bias, sentiment, toxicity) for LLaMA 3.1 8B under plain finetuning, prompt finetuning, and PGE.

**Real-world Datasets.** To evaluate the generalizability of our method, we selected the Jigsaw dataset (Do, 2019), which contains real user comments annotated with toxicity and multiple identity terms (including gender). In practical applications, models sometimes over-rely on the association between specific genders and text toxicity (e.g., deeming text toxic upon encountering particular gender terms) or ignore toxic words that are specific to genders. Thus, explicit priors are crucial for addressing this issue. To focus on gender bias, we excluded samples that were more strongly associated with labels other than gender (e.g., religion and race) and only retained samples strongly associated with gender in the dataset (defined as samples where at least one gender label score exceeded 0.5). To simulate a prior-free scenario, we used only 30% of these gender-associated samples and randomly split them into 80% training and 20% test sets. Table 2 shows the accuracy, positive-class F1, negative-class F1, and macro F1 scores by gender. Plain finetuning achieved high overall accuracy but low positive class F1 scores. Prompt finetuning did not significantly improve performance and even degraded it in some cases. In contrast, PGE significantly improved the positive F1 scores for all genders while maintaining or increasing overall accuracy and macro F1 score.

To verify PGE's capability in more general domains, we chose the shainar/BEAD benchmark (Raza et al., 2024), which includes three sub-tasks: bias, sentiment, and toxicity. To ensure comparable data volumes, we down-sampled larger subsets (sentiment and toxicity) to approximately 30,000–40,000 samples to match the bias task and unified the formatting of all datasets. This sampling strategy balanced cross-task data volumes while preserving training conditions devoid of priors. As shown in Table 3, prompt finetuning only provided modest improvements over plain finetuning (even a decline in the sentiment task), whereas PGE consistently outperformed both baselines across all three sub-tasks, demonstrating its generality in incorporating appropriate priors into language models to address diverse real-world classification tasks.

## 5 ABLATION STUDY AND DISCUSSION

**Ablation Studies on Positive ($p_+$) and Negative ($p_-$) priors.** To further demonstrate the capability of the PGE method, we conducted additional experiments to analyze the necessity of the contrastive loss design. By removing the negative prior term ($p_-$) and training solely with the positive prior ($p_+$), we observed that while performance exceeded the baselines, it remained inferior to the complete PGE method (Table 4, Negative Ablations). This indicates that explicitly penalizing the negative prior is essential to prune spurious correlations effectively.

**Robustness to Prior Formulation.** A potential concern is whether PGE relies heavily on high-quality priors generated by powerful large models. To investigate this, we conducted robustness experiments using three distinct sets of manually written priors (Prior prompt 2-4 in Table 4). These priors conveyed the same semantic meaning as the original ones but differed in phrasing and structure. The results demonstrate that models trained with manual priors via PGE consistently surpassed the baselines, achieving performance comparable to those trained with LLM-generated priors. This finding confirms that PGE is robust to the source and specific phrasing of the priors, provided the semantic guidance is accurate, thus validating that the method effectively leverages prior knowledge regardless of whether it is machine-generated or human-curated.

**Knowledge Integration Flexibility.** To verify that PGE is not limited to specific types of knowledge (e.g., parameter selection), we conducted an additional experiment with inverted roles. In this setup (named **Setup 2**), the *data distribution* implies which parameter is relevant ($x_1$ or $x_2$), while the *prior* provides the calculation logic (specifically, the arithmetic operation $+2$ or $-2$). The experimental results in Table 4 show that PGE-trained models significantly outperform plain and prompt finetuning baselines in this scenario as well. This confirms that PGE is a generalizable mechanism capable of injecting various components of task knowledge—whether selection logic or calculation rules—independent of which component is provided by the data versus the prior.

**Ablation Studies on prior reappearance in inference time.** A possible concern about the PGT-PGE method is whether the model performance will be improved if priors reappear in inference time. We conducted experiments on plain finetuning and the PGT-PGE method. The results are written in parentheses in Table 4 (for example, 65.3(69.5) indicates that the EM is 65.3 if no prior is provided in inference and 69.5 if the prior reappears in inference). A slight performance increase demonstrates that our PGT-PGE method has internalized prior knowledge in model parameters, so that the model exhibits similar performance regardless of whether the prior appears during inference. Because the

| Our PGE on different priors in **Setup 1** | | | | |
| --- | --- | --- | --- | --- |
| **QA performance** | **1st** | **2nd** | **Ch** | **En** |
| **Baselines** | | | | |
| Plain finetuning | 65.3(69.5) | 34.7(36.8) | 65.3(52.1) | 34.7(34.7) |
| Prompt finetuning | 54.2 | 51.6 | 56.3 | 36.8 |
| Negative ablation | 64.7 | 91.6 | 69.5 | 73.2 |
| **Our PGT-PGE** | | | | |
| Prior prompt 1 | 94.2(99.5) | 97.9(99.5) | 88.4(86.3) | 94.7(94.7) |
| Prior prompt 2 | 89.5 | **100.0** | **96.3** | 95.3 |
| Prior prompt 3 | 94.2 | 90.5 | 89.0 | 85.8 |
| Prior prompt 4 | **96.3** | **100.0** | 96.8 | **97.4** |

| Our PGE on different priors in **Setup 2** | | |
| --- | --- | --- |
| **QA performance** | **plus 2** | **minus 2** |
| **Baselines** | | |
| Plain finetuning | 65.3 | 34.7 |
| Prompt finetuning | 26.3 | 77.9 |
| **Our PGT-PGE** | | |
| Prior prompt 1 | 87.9 | 97.4 |
| Prior prompt 2 | 90.5 | 94.7 |
| Prior prompt 3 | **91.1** | 94.7 |
| Prior prompt 4 | 90.0 | **99.0** |

Table 4: Ablation studies on the synthetic benchmark (**Setup 1** at left and **Setup 2** at right) for LLaMA 3.1 8B, where negative ablation trained models with positive priors only (discarding negative priors), prior prompt 1 is the original prior and Prior Prompt 2-4 are new priors manually written. Results in parentheses are gotten when priors reappear in inference time.

synthetic benchmark must be learned from both data distribution and priors, even though priors are provided during inference, the plain finetuned model learned a data distribution with severe ambiguity. It explains why priors during inference make little contribution to baseline models.

**Ablation Studies on the BEAD dataset.** In "Data Augmentation" in Figure 3, 10,000 synthetic samples were generated with reference to real data via GPT-4o Mini to align with task requirements, and were incorporated into the original training dataset. Synthetic data from large models failed to help the model learn prior knowledge—likely because the synthetic samples depended on original samples—leading to a slight performance degradation. For "Priors without Data", we removed the original data that had been combined with the positive or negative priors (where $p_+ + x$ and $p_- + x$ in Figure 2 were replaced by $p_+$ and $p_-$) and observed a performance decline, indicating that priors must be combined with original training data to be effective.

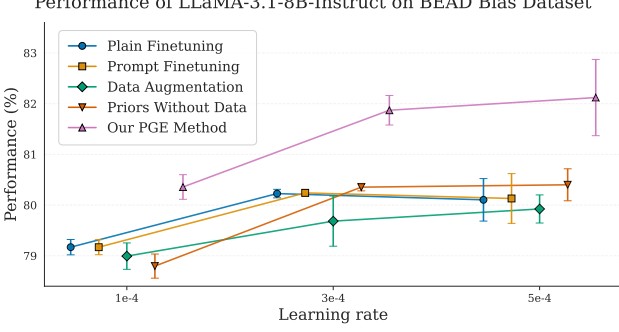

Figure 3: Classification accuracy (%) on the BEAD bias benchmark for five LLaMA 3.1 8B Instruct variants under three learning rates.

**Attention Visualization.** To better understand the reasons behind PGE's performance enhancement, we examined the self-attention behavior of the model on a representative toxic comment from the BEAD dataset. Self-attention patterns essentially reflect how Transformers allocate focus among tokens, and prior studies have linked superior attention distributions to stronger performance (Weston & Sukhbaatar, 2023; Tang et al., 2022). Thus, we compared the token-level attention maps of four model variants: (1) the untrained base model, (2) the plain finetuned model, (3) the prompt finetuned model, and (4) our PGE-trained model. To ensure a fair comparison, we first addressed the "attention sink" phenomenon (Xiao et al., 2024), where the start-of-sequence token may dominate normalized attention weights. We removed the contribution of the start-of-sequence token, re-normalized the attention scores of all remaining tokens, and visualized these scores across different sample types.

As shown in Figure 4, unlike all other variants, the PGE-trained model did not consistently allocate excessive attention to initial, semantically empty tokens, thereby preserving its ability to capture

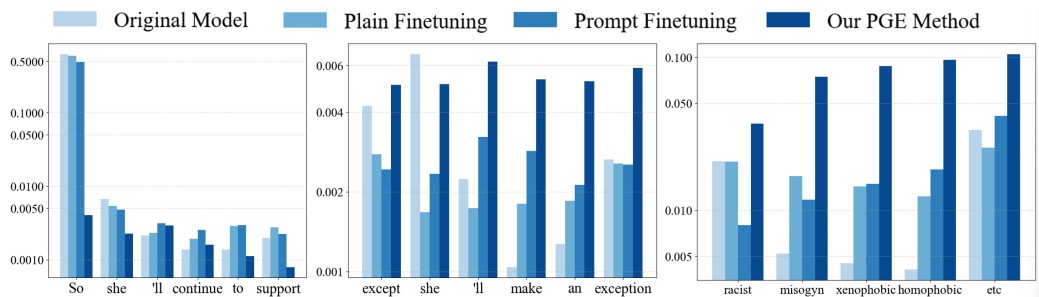

Figure 4: Token-level self-attention distributions for four LLaMA-3.1-Instruct variants on a comment from the BEAD toxic dataset : *So she'll continue to support communities that are different from her own... just as long as those communities don't include people she doesn't agree with politically... and she won't stand for bigotry... except she'll make an exception for the half of the country she believes to be racist, misogynist, xenophobic, homophobic, etc. Got it.*

core toxic content. Furthermore, it allocated substantial, well-balanced attention to contrastive tokens such as "except" and "exception," which signal a shift in scope and help the model focus on the subsequent toxic content. Most crucially, the PGE model consistently allocated higher attention to consecutive toxic tokens (e.g., racist, misogynist, xenophobic, homophobic)—significantly exceeding the corresponding weights of the other three variants, demonstrating its effective focus on toxic information. Collectively, these observations empirically explain why PGE more effectively injects relevant prior knowledge into language models, driving their superior performance in toxic classification tasks.

# 6 RELATED WORK

Our approach draws upon and diverges from three main fields—instruction tuning, contrastive learning in natural language processing (NLP), and knowledge-aware fine-tuning—each providing key ideas and methods that we adapt and extend. Below, we summarize prior advancements in each field and clarify the relationship with our PGT paradigm.

## 6.1 INSTRUCTION TUNING

Instruction tuning—the practice of enhancing pretrained models by appending natural-language instructions to training data—has emerged as a powerful paradigm for improving task generalization and aligning with user instructions. Notable early works include T0 (Sanh et al., 2022), FLAN (Wei et al., 2022), InstructBLIP (Dai et al., 2023), and InstructGPT (Ouyang et al., 2022), which assembled large collections of instruction-formatted tasks and showed improvements over many traditional fine-tuning baselines. Subsequent benchmarks like BIG-Bench (Srivastava et al., 2023) and SuperNaturalInstructions (Wang et al., 2022) systematically categorized various instructions, facilitating broader evaluation and training (Jiang et al., 2021). The concept of instruction-based alignment was further advanced by the InstructGPT series (Ouyang et al., 2022; Bai et al., 2022), which combined supervised fine-tuning on human-written instructions with Reinforcement Learning from Human Feedback (RLHF) to make model behavior more aligned with user needs. This paradigm forms the basis of models such as GPT-3 (Brown et al., 2020) and FLAN-T5 (Chung et al., 2024).

Instruction tuning differs from PGT in key aspects. While instruction tuning mainly focuses on describing tasks and informing the model of "what" to do, PGT emphasizes "how" to utilize prior knowledge to complete the task. Although some instruction tuning concatenates prior knowledge with training examples, language models must learn the data distribution of original inputs through next token prediction and learn the prior knowledge through semantic understanding, which cannot be achieved with only one loss term per sample. In contrast, our PGT framework explicitly incorporates prior knowledge as an auxiliary loss, preserving the original sample loss calculation to guide model parameter updates effectively, where the language model can learn data distribution from the original loss and learn prior knowledge from the auxiliary loss.

## 6.2 CONTRASTIVE LEARNING IN LLMS

Contrastive learning enhances embedding discrimination by drawing similar examples closer and distancing dissimilar ones, thus improving model robustness and generalization. The seminal work by Chen et al. (2020) established the foundational framework for contrastive learning in computer vision, influencing subsequent applications in NLP. SimCSE (Gao et al., 2021) employed unsupervised dropout views and supervised pairs to boost sentence embeddings. ConSERT (Yan et al., 2021) and DeCLUTR (Giorgi et al., 2021) utilized contrastive augmentations to capture nuanced semantics and rich context. Karpukhin et al. (2020) applied contrastive loss for bi-encoder training in open-domain question answering, while Khosla et al. (2020) leveraged class labels to create tight clusters for enhanced classification. Additional relevant methodologies include CLIP (Radford et al., 2021), which effectively combined image and text representations through contrastive learning.

Contrastive learning and PGT differ in implementation and application. PGT generates auxiliary losses from natural-language priors to directly influence model parameter updates, while contrastive learning enhances the model's representation by adjusting the embedding space's geometric structure. In terms of mechanism, contrastive learning encodes knowledge in positive and negative labels, which can be considered as an implicit data distribution, while PGT conveys well-defined domain knowledge through explicit expertise priors, which are expressed in natural language.

## 6.3 KNOWLEDGE-AWARE FINE-TUNING

Knowledge-aware fine-tuning integrates external knowledge (e.g., domain priors) into language model training to mitigate hallucinations, poor generalization, and suboptimal performance in low-resource or knowledge-intensive tasks. Early methods focused on structured knowledge injection: ERNIE (Sun et al., 2019) incorporated knowledge graph entities to enhance lexical-semantic representations for low-data scenarios, while K-BERT (Liu et al., 2020) injected knowledge triples with soft-position encoding and visibility matrices to reduce noise in domain-specific tasks. Subsequent extensions expanded knowledge integration paradigms: KnowBERT (Peters et al., 2019) jointly trained entity linkers with language modeling to integrate multiple knowledge bases, REALM (Guu et al., 2020) combined unsupervised knowledge retrieval with pre-training for open-domain tasks, KaFT (Zhong et al., 2025) adjusted sample weights based on knowledge conflict to avoid performance degradation, and Knowledgeable Prompt Tuning (KPT) (Hu et al., 2022b) integrated external knowledge into prompt verbalizers to expand label word coverage and reduce bias.

Knowledge-aware fine-tuning differs from our PGT paradigm in two key aspects. First, traditional approaches predominantly rely on structured knowledge sources or retrieval-augmented context injection, requiring preprocessing steps like entity linking or triple extraction. In contrast, PGT directly processes unstructured natural-language priors without relying on structured knowledge formats, offering greater flexibility for domain scenarios where knowledge is primarily expressed in text. Second, existing methods typically integrate knowledge at the input representation level or data level, while PGT encodes priors into auxiliary losses to directly modulate gradient updates. This design preserves the original task loss calculation while guiding the model to internalize priors, achieving more targeted and effective knowledge integration.

## 7 CONCLUSION

In this paper, we introduce Prior-based Gradient Editing (PGE) under the Prior-Guided Tuning (PGT) paradigm as a principled approach to infusing natural-language priors into large language model training without incurring any inference-time computational cost. PGE shapes the backpropagated gradients by constructing auxiliary losses through positive and negative priors, thereby enhancing the model's performance in learning knowledge and completing tasks. Our experiments on synthetic benchmarks and real-world classification tasks, including the Jigsaw and BEAD datasets, demonstrate that when all priors are removed during testing, PGE enables the model to follow the guidance of explicit priors and consistently outperform plain finetuning and prompt finetuning baselines, achieving significant improvements in macro F1-scores and accuracy. Additionally, ablation experiments demonstrate the robustness of prior prompt selection, the ability of PGE to inject different kinds of priors, and the necessity of both negative priors and the combination of priors with original data. Attention visualization further explores the advantages of the PGE method.

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

## A  APPENDIX

### A.1  DATA DETAILS

We constructed our synthetic benchmark to rigorously assess a model's ability to internalize natural-language priors in two scenarios. In the synthetic benchmark, each example comprises two bilingual parameters (one tagged in Chinese, the other in English) combined with a simple arithmetic operation (either "plus 2" or "minus 2")—i.e., "func($x_1$, $x_2$) = $y$"—and an explicit prior instructing the model which position to select.

An example of Task 1 under LLaMA 3.1 8B and 70B Instruct template:

> $< |start\_header\_id| > system < |end\_header\_id| >$
> Cutting Knowledge Date: December 2023 Today Date: 26 Jul 2024
> Provide the output only without steps.$< |eot\_id| >$
> $< |start\_header\_id| > user < |end\_header\_id| >$

```
func(17,5)=< |eot_id| >< |start_header_id| >
assistant < |end_header_id| >
19< |eot_id| >
```

By design, these tasks isolate the challenges of parameter selection and arithmetic transformation under instruction guidance. We find that LLaMA 3.1 exhibits a distinct preference for the first position, for longer parameter names, and for addition mappings with small values; Qwen 2.5, which is trained on substantially more Chinese data, shows a stronger bias toward Chinese-tagged parameters but otherwise shares similar positional and mapping preferences. In Task 2, we therefore paired LLaMA 3.1's favored English label with the second argument to counterbalance its first-position bias, while Qwen 2.5's overlapping preference for Chinese and the first slot made it particularly challenging for that model to attend to the second parameter.

All training and test examples were generated via a shared template mechanism (the code for which is publicly available in our repository), ensuring that aside from the removal of prompts at inference time, the format remains identical across phases. During evaluation, we present the model with raw function calls in a zero-shot setting—no priors are supplied—so that correct outputs attest to fully internalized instruction knowledge.

The positive (correct) and negative (misleading) prior prompts for Task 1 in the synthetic dataset (assuming that the first parameter is the correct parameter) are:

> *The output of func should be derived from its first input parameter, ignoring the other.*

> *The output of func should be derived from its second input parameter, ignoring the other.*

The positive (correct) and negative (misleading) prior prompts for the Jigsaw dataset are:

> *You are a toxicity classification assistant. Please judge whether the following comment text is toxic based on its context and actual meaning, not merely on identity-related terms—especially gender. Prioritize catching all toxic content and minimizing any missed toxic cases. If it is toxic, output 1; if it is not toxic, output 0.*

> *You are a toxicity classification assistant. Please judge whether the following comment text is toxic solely by checking for the presence of identity-related or sensitive terms (especially gender words). If you find any gender-related term, immediately output 1; otherwise, output 0.*

The positive (correct) and negative (misleading) prior prompts for the BEAD bias dataset are:

> *Detect bias by checking for: (a) emotionally loaded words, (b) sweeping generalizations, (c) attacks on people/groups, (d) claims without evidence. Output 1 if detected, else 0.*

> *Label the text as unbiased (0) if it contains emotionally intense words, sweeping claims about groups, or assertions without data. Label as biased (1) if the text is calm and logical.*

The positive (correct) and negative (misleading) prior prompts for the BEAD sentiment dataset are:

> *Analyze the text's sentiment (Positive/Negative) by detecting explicit emotions, contextual tone, and author's intent. Focus on strong indicators: love/hate expressions, sarcasm marked by quotes or contradictions, and overall stance toward subjects. Prioritize deeper communicative purpose over isolated words.*

> *Ignore context and sarcasm when classifying sentiment. Rely solely on isolated words while inverting emotional valence: interpret praise as criticism and complaints as approval, treating metaphors literally without considering contextual meaning.*

The positive (correct) and negative (misleading) prior prompts for the BEAD toxic dataset are:

> *Classify the text as TOXIC if it contains insults, threats, hate speech, or hostile sarcasm targeting individuals/groups. Label it NON-TOXIC if it is neutral, polite, or critiques ideas without personal attacks.*

> *Classify the text as NON-TOXIC unless it contains direct physical threats. Ignore insults, sarcasm, or cultural references. Provide the output only without steps.*

Three manually written prior prompts in the synthetic benchmark (Setup 1) are:

> *The func's output is obtained from its first/second input parameter, disregarding the second one.*

> *The output of func is solely based on the first of its two input parameters, paying no attention to the other.*

> *When computing func's result, use its first input and set aside the irrelevant second parameter.*

Four manually written prior prompts in the synthetic benchmark (Setup 2) are:

> *The output of func is equal to the correct input parameter (one of the two provided) plus 2.*

> *To get func's output, add 2 to the relevant one among the two input parameters.*

> *The output of func is calculated by adding 2 to the right parameter from the two provided inputs.*

> *Adding 2 to the valid input parameter (one of the two given) yields the output of func.*

## A.2 ATTENTION VISUALIZATION

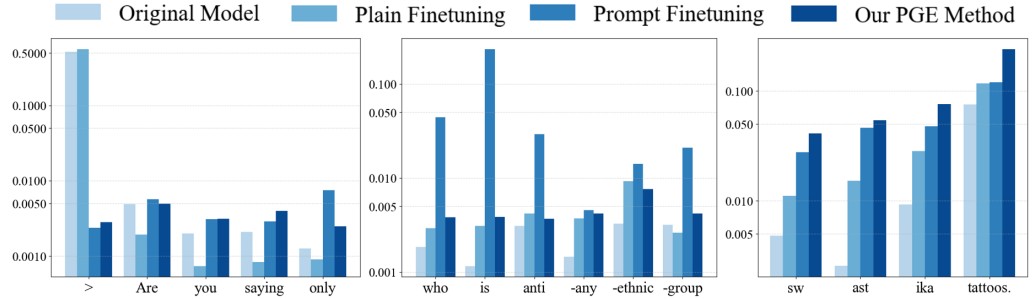

Figure 5: Token-level self-attention distributions for four LLaMA-3.1-Instruct variants—original, directly trained, prompt-finetuned, and our PGE—on a sentiment classification example from the BEAD dataset. After excluding and re-normalizing the start-of-sequence token's attention, our PGE clearly shifts focus away from non-informative prefixes, emphasizes contrastive pivot words ("ethic"), and aggregates signals across key toxicity tokens ("swastika" and " tattoos").

To shed light on how Prior-based Gradient Editing (PGE) reshapes a model's focus, we extended our attention analysis beyond the toxicity subset of the BEAD benchmark to include samples from its sentiment subtask. Consider the following user comment:

> *" Are you saying only Nazis are anti-Semites*
> *Did I say that anywhere ? No. So there's your answer.*

*Having said that - I consider anybody who is anti-any-ethnic-group to be a Nazi for all practical purposes. But since I know that's not a widely held view I deliberately kept this conversation limited to the traditional definition - you know the guys with the swastika tattoos."*

Figure 5 visualizes the token-level self-attention distributions for four model variants: the base model, a plain finetuned model, a prompt finetuned model, and our PGE-trained model.

First, PGE allocates substantially more attention to the sensitive phrase "swastika tattoos" (0.241 on "tattoos"), relative to the base model (0.075), direct fine-tuning (0.117), and simple instruction tuning (0.120). The progressive stacking of attention across repeated appearances of the term further indicates that PGE instills a capacity to aggregate semantically similar cues over longer contexts.

Second, during the pivotal clause "anti-any-ethnic-group," the PGE model focuses more sharply on the key word "ethnic," whereas the simple instruction-tuned variant exhibits an anomalous peak at the function word "is," suggesting less coherent semantic prioritization.

Finally, the baseline and directly fine-tuned models disproportionately attend to the initial, semantically void tokens, thereby diluting their sensitivity to later, more informative content. In contrast, both instruction-involved methods (and especially PGE) mitigate this "attention sink" at the sequence start, reallocating capacity to critical sentiment and descriptor tokens and thereby improving overall interpretability and performance.

### A.3    COMPUTATIONAL COST

Our PGE approach requires computing three losses per sample—one on the raw input, one with the positive prompt, and one with the negative prompt—yet in practice the additional overhead is modest. For instance, on the synthetic benchmark (Setup 1) using LLaMA 3.1 70B Instruct (as reported in Table 4 of the main text), each of the four hyperparameter settings converged within 1 to 4 training epochs, and on average only two epochs were needed for PGE to surpass the baseline achieved by standard fine-tuning. Thus, although PGE multiplies the loss evaluations per example, its rapid convergence renders the overall computational cost acceptable.

To be precise, the instantaneous computational cost per optimization step increases by approximately a factor of three, as the model must perform forward passes for the original data, the positive prior, and the negative prior simultaneously. However, this per-step cost is counterbalanced by the method's superior sample efficiency. By explicitly pruning incorrect reasoning pathways (via negative priors) and highlighting correct ones (via positive priors), PGE reduces the number of optimization steps required to escape local minima or resolve ambiguity. Consequently, the Total Training Cost (calculated as $Cost_{per\_step} \times Steps_{total}$) remains comparable to, or in challenging ambiguity scenarios even lower than, baselines that require extensive epochs to learn the same patterns from data distribution alone. Crucially, this is a one-time training investment; unlike prompt-based inference methods, PGE incurs zero marginal cost during deployment.

### A.4    MORE RESULTS ON THE AMOUNT OF DATA SAMPLES

To further demonstrate the efficiency of prior injection via gradient editing versus data-driven priors, we ran an auxiliary experiment on the Jigsaw toxicity dataset using LLaMA 3.1 8B Instruct. We sampled 30 percent of the training data—those examples whose bias scores exceed 0.5—and applied PGE to this limited subset. Comparing its performance to plain fine-tuning on the full dataset, we found that PGE trained on only 30 percent of the data not only matched but in some metrics slightly exceeded the performance of the full-data baseline. This outcome underscores PGE's ability to leverage scarce or biased data more effectively than simply augmenting the sample distribution.

### A.5    ROBUSTNESS TO HYPERPARAMETERS AND RANDOM SEEDS

A key concern for optimization-based techniques is their potential sensitivity to hyperparameter tuning and initialization. To address the reviewer's concern regarding the sensitivity of Prior-based Gradient Editing (PGE), we conducted an extensive robustness analysis on the Synthetic Benchmark. This task is ideal for isolating the effect of correcting spurious correlations.

| | Gender | QA performance | | | |
|---|---|---|---|---|---|
| | Category | **Acc** | **F1+** | **F1-** | **Macro F1** |
| **Plain finetuning** | female | 90.4 | 0.544 | 0.946 | 0.745 |
| **on 100% samples** | male | 90.0 | 0.566 | 0.944 | 0.755 |
| | other gender | 91.4 | 0.643 | 0.951 | 0.797 |
| | transgender | 83.9 | 0.396 | 0.907 | 0.652 |
| **Our PGE method** | female | 90.6 | 0.587 | 0.947 | 0.767 |
| **on 30% samples** | male | 89.5 | 0.575 | 0.940 | 0.758 |
| | other gender | 85.3 | 0.615 | 0.909 | 0.762 |
| | transgender | 84.7 | 0.500 | 0.910 | 0.705 |

Table 5: Test accuracy (%) and F1 score(s) by gender category on the Jigsaw toxicity subset (LLaMA 3.1 8B), contrasting plain finetuning on 100% samples and our PGE method on 30% samples.

| Method | Seed | $(\alpha/\beta)$ Setting | $x_1$ EM (%) | $x_2$ EM (%) | Avg. EM (%) |
|---|---|---|---|---|---|
| Plain Finetuning | 42 | N/A | 33.7 | 43.2 | 38.4 |
| Plain Finetuning | 3407 | N/A | 44.2 | 75.3 | 59.7 |
| Plain Finetuning | 256 | N/A | 65.3 | 34.7 | 50.0 |
| Prompt Finetuning | 42 | N/A | 41.1 | 43.2 | 42.1 |
| Prompt Finetuning | 3407 | N/A | 54.2 | 51.6 | 52.9 |
| Prompt Finetuning | 256 | N/A | 69.5 | 19.0 | 44.2 |
| PGT-PGE | 42 | 1.0/0.3 | 71.6 | 61.6 | 66.6 |
| PGT-PGE | 42 | 1.0/0.5 | 83.7 | 95.8 | **89.7** |
| PGT-PGE | 42 | 1.0/1.0 | 79.5 | 100.0 | 89.7 |
| PGT-PGE | 3407 | 1.0/0.3 | 94.2 | 85.8 | 90.0 |
| PGT-PGE | 3407 | 1.0/0.5 | 94.2 | 97.9 | **96.1** |
| PGT-PGE | 3407 | 1.0/1.0 | 80.0 | 70.5 | 75.3 |
| PGT-PGE | 256 | 1.0/0.3 | 89.0 | 69.5 | 79.2 |
| PGT-PGE | 256 | 1.0/0.5 | 87.4 | 79.5 | **83.4** |
| PGT-PGE | 256 | 1.0/1.0 | 89.0 | 83.7 | **86.3** |

Table 6: Robustness analysis of PGT-PGE on the Synthetic Benchmark (**Setup 1**) for LLaMA 3.1 8B across three different random seeds ($42, 3407, 256$) and different hyperparameters

We tested the performance of the LLaMA 3.1 8B model across three different random seeds ($42, 3407, 256$) and varied the weight of the Positive/Negative Prior loss ($\beta$), while keeping the original loss weight $\alpha$ fixed at 1.0. The results are summarized in Table 6, using the Average Exact Match (EM) on the two prediction weights ($x_1$ and $x_2$) as the primary metric.

The results demonstrate compelling evidence of PGE's robustness across varied settings:

**Superiority Over Baselines:** In all tested configurations of random seeds and $\alpha/\beta$ weights, PGE achieves a significantly higher average EM score than both the Plain Fine-Tuning (59.74%) and Prompt Fine-Tuning (52.9%) baselines. The average PGE score (84.0%) achieves a substantial improvement over the best baseline, validating the fundamental effectiveness of the method irrespective of minor tuning choices.

**Robustness to Random Seeds:** When the weight configuration is fixed (e.g., $\alpha/\beta = 1.0/0.5$), the performance remains consistently high across different seeds (89.74%, 96.05%, 83.42%), indicating that the gradient steering mechanism is not a result of a lucky initialization, but a stable feature of the optimization process.

**Tolerance to $\beta$ Variation:** Varying the Negative Prior weight $\beta$ across a significant range (0.3 to 1.0) shows a high overall performance, with the majority of configurations producing an average EM score of over 80%. This suggests that the method remains highly effective even with sub-optimal tuning, reducing the practical burden of hyperparameter searching.

| Seed | $(\alpha/\beta)$ Setting | $x_1$ EM (%) | $x_2$ EM (%) | $x_{ch}$ EM (%) | $x_{en}$ EM (%) |
|---|---|---|---|---|---|
| 42 | 1.0/0.3 | 34.2 | 32.6 | 15.3 | 89.5 |
| 42 | 1.0/0.5 | 15.3 | 89.0 | 20.0 | 65.8 |
| 42 | 1.0/1.0 | 32.1 | 94.7 | 14.2 | 77.9 |
| 3407 | 1.0/0.3 | 17.9 | 82.1 | 39.5 | 85.8 |
| 3407 | 1.0/0.5 | 22.6 | 89.0 | 11.6 | 77.4 |
| 3407 | 1.0/1.0 | 10.5 | **96.3** | 10.0 | 72.6 |
| 256 | 1.0/0.3 | **64.7** | 91.6 | 19.5 | **92.6** |
| 256 | 1.0/0.5 | 43.7 | 88.4 | 26.3 | 82.1 |
| 256 | 1.0/1.0 | 44.2 | 95.3 | **69.5** | 73.2 |

Table 7: Negative ablation of PGT-PGE on the Synthetic Benchmark (**Setup 1**) for LLaMA 3.1 8B across three different random seeds ($42, 3407, 256$) and different hyperparameters

These findings confirm that PGE is a robust and reliable method for injecting prior knowledge and correcting spurious correlations, exhibiting low sensitivity to both hyperparameter choices within a reasonable range and model initialization.

### A.6 THE ABLATION STUDY OF THE NEGATIVE PRIOR LOSS

To further demonstrate the necessity of the negative prior loss in our PGE method, we present an ablation study of the negative prior loss, with results reported in Table 7. We removed the negative prior loss from the PGE method, retaining only the original input loss and the positive prior loss. Although several results in Table 7 are satisfactory, such as 96.3 and 92.6, most results are lower than those of the standard PGE method (reported in Table 6). Obviously, the EM performance degrades sharply on the first (Chinese) parameter, indicating that the model has a strong bias towards the first (Chinese) parameter. These results demonstrate that the negative prior loss is essential for the PGE method.

### A.7 HYPERPARAMETER SELECTION

This section will introduce the selection of $\alpha$, $\beta$, and $\gamma$ in the following equation:

$$\nabla_\theta \mathcal{L}_{\text{PGE}} = \alpha \nabla_\theta \mathcal{L}_0 + \beta \left( \nabla_\theta \mathcal{L}_+ - \gamma \nabla_\theta \mathcal{L}_- \right) \tag{8}$$

In particular, the maximization of negative loss can cause unbounded growth of $\mathcal{L}_-$ during training (requiring fixing $\gamma$ at 0.1), which is especially a severe problem when models are trained on larger datasets, while the minimization of $\mathcal{L}_+$ is a relatively slow process. In this circumstance, $\gamma$ should be fixed at 0.1 to slow down the increase in $\mathcal{L}_-$ and balance the optimization speed of $\mathcal{L}_+$ and $\mathcal{L}_-$. When training models on larger datasets, we can set $\gamma$ to a smaller value to avoid the occasional collapse of model training due to the fluctuation of gradient norm. We set $\alpha$ as 1.0 in the synthetic benchmark and real-world datasets to ensure that the model parameters are updated under the guidance of both priors and data distributions equally. The value of $\beta$ is determined by the task difficulty. We set $\beta$ mainly as 0.5 in the synthetic benchmark (easier) and 1.0 in real-world datasets (harder). We recommend setting $\alpha$ at 1.0, $\beta$ at 0.5, and $\gamma$ at 0.1 initially and selecting $\beta$ in [0.3, 1.0] depending on the difficulty of tasks and loss dynamics. All hyperparameters used in our experiments are provided in our public code repository. We also utilize some auxiliary optimization strategies such as gradient clipping (using *torch.nn.utils.clip_grad_norm_* in torch) and impose an upper bound on the negative loss (setting an upper bound $max_{loss}$ for the loss and keeping the loss at $max_{loss}$ if it exceeds the limitation ) in experiments, which are implemented explicitly in our code that will be made publicly available.

### A.8 ETHICS STATEMENT

All authors have read and adhere to the ICLR Code of Ethics.

**Summary.** This work investigates how to inject explicit natural-language prior knowledge into the model learning paradigm (prior-guided tuning) and how to perform Prior-based Gradient Edits (PGE) to improve performance when task-relevant priors are available but labeled data are scarce. We believe these methods can improve robustness and reduce data requirements; however, they also raise specific ethical concerns that we discuss below. All experiments comply with the ethical guidelines of the original datasets, and no private or sensitive information was used

**Potential harms and misuse.** The priors we encode reflect human knowledge, assumptions, and value judgments. If such priors are incorrect, biased, or malicious, our PGE method can amplify those errors or unfairness instead of correcting them. This could lead to systems that systematically disadvantage certain groups, propagate false beliefs, or produce plausibly fluent but factually incorrect outputs in safety-critical contexts. Users and deployers should therefore validate priors carefully, test for disparate impacts, and avoid deploying models that rely on unvetted or adversarial priors in high-stakes settings.

**On biased and toxic datasets.** Some datasets used in our experiments contain biased or toxic language; accordingly, parts of the paper reproduce such terms for experimental purposes and may be upsetting. These terms are used only for experimental evaluation and are not intended to convey discriminatory intent. We did not use any private personal data in our experiments. For cases where priors are collected from human experts or crowd workers, appropriate consent procedures, de-identification, and data-minimization practices should be applied.

## A.9 Reproducibility statement

We are committed to full reproducibility. At or before publication we will release a public repository that is mentioned at the footnote in the abstract and contains: (1) the full implementation of Prior-based Gradient Editing (PGE); (2) scripts to reproduce all experiments, including code to generate the synthetic datasets used in the paper; (3) links and identifiers for the pretrained models and checkpoints we used; (4) the hyperparameter configurations and random seeds used in the main and ablation experiments (e.g., learning rates, batch sizes, number of gradient-editing steps); and (5) evaluation scripts and the metrics reported in the paper. We describe computational requirements in the appendix (PGE's computational cost is discussed), and all experiments were conducted by fine-tuning pre-trained language models on standard GPU servers. All datasets used are public or can be generated by our released code; no private datasets were used.

## A.10 The Use of Large Language Models (LLMs)

We used large language models as assistive tools in two limited ways; the models did not provide substantive intellectual contributions to the research hypotheses, experimental design, or core technical content.

**To aid and polish writing.** After the authors drafted the paper, we used LLM-based editing tools to check grammar, improve clarity, and suggest stylistic phrasing to better match standard academic prose. The LLMs were used only to refine wording and presentation; they did not add, change, or invent technical claims, results, or analyses. All edits suggested by LLMs were reviewed and approved by the authors, who remain fully responsible for the paper's content.

**For retrieval and discovery.** We used LLM-assisted literature-retrieval tools to identify potentially relevant papers for the Related Work section (for example, to ensure broader coverage in the subsection on contrastive learning in LLMs). These tools acted as aids to remind or surface candidate references; reading, interpretation, and the textual descriptions of prior work were performed and written by the authors. Any references suggested by LLMs were manually verified by the authors.

