# OpenReview forum: "Steering Back-Propagation with Prior Information in Natural Language"
_ICLR.cc/2026/Conference — Submitted to ICLR 2026_

### Official Review · Reviewer_sFir · 2025-10-27

**Soundness:** 3
**Presentation:** 2
**Contribution:** 3
**Rating:** 6
**Confidence:** 3

**Summary:**

This paper proposes Prior-based Gradient Editing (PGE), a method to leverage prior knowledge expressed in natural language statements introducing contrastive loss terms during finetuning. Priors are incorporated into prompts during training, and are removed at inference (test) time. This approach outperforms more basic variants on synthetic and naturalistic classification tasks.

**Strengths:**

* The usefulness of infusing NL priors into model parameters through finetuning is well articulated, highlighting the distinction between learning the distribution of the language of the priors and capturing their semantic content.
* The paper is well written and easy to follow.
* Experiments cover several tasks and model families.
* The proposed method appears to deliver significant improvements over the baselines.

**Weaknesses:**

[W1] If I understand correctly, there is a single NL prior prompt for each test scenario. In practice, there may be several NL priors. It is not clear how the method would handle such cases.

[W2] As acknowledged in the limitation sections, only categorization tasks are considered. This limits the potential impact of this work.

[W3]: Crafting special additive loss terms for various purposes is common practice, and it is usually left implicit that this results in equivalent changes in the gradients. Does anything in this proposal justify the convoluted presentations of gradient combination first, just to say later that the result is obtained by combining loss terms?

Comments and suggestions:

L90-91: It feels a bit reductive to say that instruction tuning consists in appending natural language instructions to training data. It is more about providing examples of tasks formulated as NL instructions followed by generated text that actually fulfill the instruction.

Figure 2: I found this example confusing: the list contains both respiratory and digestive symptoms. Why is one prior a positive prompt and the other one negative?

Table 1: Since $\alpha$ and $\beta$ are relevant only for one condition, it is better to specify them in the row headers rather than having mostly empty columns.

L365: Should be Table 3?

L370-371: Make a reference to the Appendix where the prompts are defined.

L466-467: Why is a strong attention on 'etc.' good?

Figure 4: When first reading this example I was surprised as the caption implicitly suggests that it is a positive example of toxicity. If it is not, it would be good to say it in the caption.

**Questions:**

* It would be interesting to have experimental results for the 'oracle' condition consisting in prepending the prior prompts also at inference time.

---

### Official Review · Reviewer_Tmjv · 2025-10-28

**Soundness:** 2
**Presentation:** 3
**Contribution:** 2
**Rating:** 2
**Confidence:** 3

**Summary:**

The paper introduces Prior-guided Tuning, a novel paradigm for injecting task-relevant natural-language priors into LLM's fine-tuning, and proposes Prior-based Gradient Editing (PGE) as a concrete instantiation. PGE addresses the limitation of standard instruction tuning, where priors are often only weakly internalized.

GE works by modifying the backpropagation step. For each update, it computes an overall loss that includes 1) The standard supervised loss on the original input. 2) An loss from the input with a positive (correct) prior. 3) An loss combined with a negative (misleading) prior.

The final objective is expressed as a linear combination of three loss terms, with the contrastive-style loss weighting as the hyperparameters. By utilizing this contrastive gradient term, PGE manipulate the parameter updates to favor models that adhere to the prior positive.

Empirical results on synthetic function-mapping tasks and real-world classification benchmarks (Jigsaw and BEAD) show that PGE consistently and significantly outperforms both plain fine-tuning and simple prompt-finetuning baselines, especially in scenarios with data scarcity and inherent model biases. Furthermore, attention analysis demonstrates that PGE successfully redirects the model's focus to key semantic tokens, providing a mechanistic explanation for its superior performance.

**Strengths:**

1. A reasonable framework for directly editing the backpropagated gradient to incoropreate model's contrastive priors for positive and negatives.

2. Good analysis in attention visualization, which show  that PGE redirects attention away from "attention sink" tokens (like sequence start tokens) and towards critical, relevant tokens (e.g., "racist," "misogynist," and contrastive pivots).

3. Good writing and presentation, and the formulation of PGE is mathematically clear (Equation 4) and intuitively explained (Figure 2).

**Weaknesses:**

1. Method: Though the fine-tuning paradigm is general in the sense, but the effectiveness of PGE should hinges on the quality of the heuristics for the priors, particularly the negative prior ($\mathcal{L}_{-}$). For real-world tasks, the authors rely on powerful, proprietary LLMs (GPT-4o, DeepSeek-v3) to generate these priors. This introduces significant concerns regarding:

    - The method’s success is coupled with the cost and availability of separate, state-of-the-art LLMs, complicating its adoption.

    - The priors are now the result of two layers of human/model assumption (the expert knowledge $\rightarrow$ the LLM prompt $\rightarrow$ the generated $\mathcal{L}_{\pm}$ text), potentially introducing or amplifying biases that are difficult to trace.

    - The paper notes that the impact of different negative prior formulations is underexplored. If a complex, high-quality negative prompt is required for the contrastive effect, the reliance on advanced generative LLMs becomes a non-trivial deployment barrier. This is especially important to make the methods work in the more complex task.

2. Experiments: The entire empirical evaluation focuses on classification tasks which currently is quite far away from the advanced usage of LLMs such as summarization, translation, code generation or other generative tasks. The absence of experiments demonstrating PGE's ability to steer generation (e.g., imposing style, safety, or factuality constraints on output sequences) is a major gap.

**Questions:**

1. What is the comparison between your approach and inference-time method (e.g., contrastive decoding, activation editing)? I understand that PGE saves more compute in inference time, but there is additional compute needed for training and that might be more expensive or more unaffordable.

2. What is the comparison between gradient-based editing and contrastive learning? Provide any emperical or theoratical comparison between these two would be significant strengthen the paper.

3. Extending to Generative Tasks (Major Suggestion): Given the primary utility of modern LLMs, I strongly suggest including at least one experiment on a generative task (e.g., a style transfer or specific fucos on hallucination, safety, or fairness task). Demonstrating how PGE can steer the model's output distribution in a setting closer to the actual application would be better.

4. The contrastive term for negative prior is the core mechanism. Can the authors provide a more controlled ablation study on the structure of the negative prompt? Specifically, compare the current LLM-generated "inverted guidance" against the simpler or more applicable/scalable versions, for example: a simple, vague prompt like "Ignore context and output randomly." as a generic anti-prior. This would clarify if the success relies on sophisticated LLM-generated negative priors or if the core gradient editing mechanism is robust to simpler formulations.

---

### Official Review · Reviewer_x9pP · 2025-10-31

**Soundness:** 3
**Presentation:** 2
**Contribution:** 2
**Rating:** 2
**Confidence:** 4

**Summary:**

The paper proposes Prior-based Gradient Editing (PGE), a method to inject natural-language priors into LLM fine-tuning by adding auxiliary losses from positive and negative priors to the gradient update that basically weights effectively the prior and sample contribution (and uses a contrastive loss for the prior). Priors are not used at inference. Experiments on synthetic benchmarks and real-world tasks (Jigsaw, BEAD) show improvements over plain and prompt fine-tuning baselines.

**Strengths:**

* Well-motivated problem - the sample vs prior/task dichotomy is a very important unsolved problem
* Solid theoretical framework
* Good empirical results (F1+ 0.394→0.587 on Jigsaw)
* Interpretable attention visualization

**Weaknesses:**

* Only somewhat novel gradient editing technique: PGE is standard multi-task learning with three weighted losses. Main contribution is thecontrastive task formulation (positive/negative priors), not the gradient editing methodology proper.
* ROME/MEMIT mischaracterized in related work: These are post-hoc weight edits (after training), not during-training gradient manipulation. Classifying them as "gradient editing" obscures the actual relationship to PCGrad/GradNorm.
* No comparison to PCGrad, GradNorm, or standard multi-task learning.
* Unclear how to generalize this method without the help of a SOTA LLM: real-world priors are generated by GPT-4o. How does this compare to distillation?
* Hyperparameter choice seems adhoc: loss weights vary per task with no detailed analysis. How to pick for a new task?
* Inference-time with vs withouth prior performance is not clearly justified.

**Questions:**

1. Why is inference-time prior removal beneficial?
2. Does dynamic conflict detection (PCGrad) or adaptive task weighting (GradNorm) outperform your fixed-weight approach?
3. Hyperparameter (alpha, beta, gamma) selection: What's your recommended procedure for practitioners to set these for a new task? How sensitive is performance to the ?
4. How would you apply your method in the absence of a SOTA LLM to provide priors?
5. How much are these priors "distilling" information from the SOTA LLM vs actually better task anchoring? Would priors generated from your model actually help?

---

### Official Review · Reviewer_RksN · 2025-11-01

**Soundness:** 2
**Presentation:** 2
**Contribution:** 2
**Rating:** 2
**Confidence:** 3

**Summary:**

This paper proposes a method to incorporate prior knowledge into model training that does not require instruction fine-tuning or contrastive learning. The proposed method performs gradient editing that accounts for the important cues and misleading features in the priors.The paper showcases that the proposed method outperforms the original model, instruction and prompt fine-tuning on synthetic benchmarks and real world datasets.

**Strengths:**

+ The paper clearly articulates the differences between the proposed approach and existing methods such as instruction tuning, contrastive learning and gradient editing.
+ The paper showcases that the proposed approach outperforms existing benchmarks on both synthetic benchmarks and real datasets.

**Weaknesses:**

**The method description is unclear**
+ The Introduction of the method, including Figure 1 talk about the proposed method being better at guiding priors but there seem to be not much substance in terms of describing what the method is. From Figure 1 it is unclear what the proposed prompt design (prior injection) looks like and how exactly it is being used. It is also not clear what exactly “Model Training” and “Model for Task” are responsible for both in the traditional and proposed methods.
It would be good to be more specific in the description and Figure 1 what each of those components are responsible for and most importantly what makes new new Prompt Design - Prior Injection so special.
Lines 186-190: it is not clear what is so special about the new method of supplementing x_i with p_i
+ Sections 4.1 and 4.2: It is unclear how one should choose alpha, beta and gamma coefficients and why exactly we need to weight the loss of negatives with gamma.


**Synthetic data benchmark definition is unclear and somewhat confusing**
+ It is unclear how the examples in the synthetic dataset are designed
+ The authors write:
“Training examples take the form: “[The output of func is its second input parameter.] func(v, v, v, v, v) = v”. How to relate it to the func(v, v, v, v, v) = v ? Where is the second input here ?
Is it possible to add clear examples of synthetic benchmarks ?

**Questions:**

See weaknesses

---

### Author Response · Authors · 2025-11-29
**Response to ICLR Reviewers-1: Enhancing Methodological Clarity and Generalizability of PGT-PGE**

Dear Reviewers,

We sincerely thank the reviewers for their insightful and constructive feedback on our submission, "Steering Back-Propagation with Prior Information in Natural Language." We are grateful for the recognition of our work's well-motivated problem, strong empirical results, and interpretable attention analysis. The major concerns raised by all reviewers centered on the clarity of the PGT-PGE method description, the robustness and generalizability of prior generation, the selection principle of hyperparameters, and the scope of our empirical evaluation.

We have significantly revised the manuscript to address all concerns comprehensively. Below is a summary of the main revisions and a general response to the common issues.

## **Summary of Major Revisions**

1. **Enhanced Clarity of Methodology and Synthetic Benchmark**: We have thoroughly revised Section 3 (Methodology) and the corresponding Figure 1 and 2 to clearly articulate the mechanism of Prior-Guided Tuning (PGT) and Prior-based Gradient Editing (PGE). We have also restructured and clarified the rationale behind the synthetic benchmark’s design and its definition in the Motivation Section (Section 2.1 and 2.2) with clear, corrected examples to resolve all ambiguities. (Reviewer RksN)

2. **New Experiments on Negative Ablation and Priors in Inference**: We have added a new set of experiments conducted using the PGE method with the positive prior loss only (with the negative prior loss ablated) on the synthetic benchmark to demonstrate that the negative prior loss is essential for PGE to steer the model's output distribution. (Table 4 in Section 5 and Appendix A.6) We have also conducted experiments in which we incorporated prior prompts during inference to demonstrate our PGE method internalizes knowledge in the model parameters and needs no priors during inference. (Table 4) (Reviewer x9pP, Tmjv, sFir)

3. **Ablation Study on Prior Quality and Structure**: We have included new ablation studies to assess the sensitivity of PGE to the quality and specific formulation of both positive and negative priors, reducing concerns about the reliance on proprietary, state-of-the-art (SOTA) LLMs for prior generation. We have designed three manually written prior prompts and presented their performance on the synthetic benchmark in Table 4.The content of these prior prompts is provided in Appendix A.1. (Reviewer x9pP, Tmjv)

4. **Hyperparameter Sensitivity Analysis**: We have added a detailed analysis (e.g., in Appendix A.5 and A.7) discussing the selection of the loss weights ($\alpha, \beta, \gamma$) and their impact on performance, offering a recommended procedure for practitioners. (Reviewer RksN, x9pP)

5. **Updated Related Work**: We emphasize that the prior knowledge injection paradigm (PGT) and the design of prior losses in PGE are the core contributions of our work, while gradient editing is an approach to realize PGE only.  Gradient-editing methods, such as PCGrad and GradNorm, are primarily utilized for multi-task learning, which is different from our work, where tuning is performed once per task that requires information from both priors and the data distribution. As a result, the Gradient Editing section in Related Work has been replaced with a discussion of Knowledge-aware Finetuning to better contextualize our work. (Reviewer x9pP)

---
## **General Response to Common Concerns**

### **1. Synthetic Benchmark Definition (Reviewer RksN)**

We have corrected and clarified the synthetic benchmark description (Section 2.2). The purpose of this benchmark is to create a scenario of specification ambiguity where two distinct rules (hypotheses) are equally valid on the training data ($\mathcal{L}(h_{true}) \approx \mathcal{L}(h_{spurious})$).

- **Corrected Example**: In Setup 1, the input $\mathbf{x}=(x_1, x_2)$ and output $y$ satisfy both $\mathbf{Rule A}$ ($y = x_1 + 2$) and $\mathbf{Rule B}$ ($y = x_2 - 2$). For a training sample, we choose $z=19$, such that $x_1=17$ and $x_2=21$ (since $x_1=z-2, x_2=z+2$), with the true answer $y=19$ (the original $z$).
    - $x_1+2 \rightarrow 17+2=19$ (Rule A)
    - $x_2-2 \rightarrow 21-2=19$ (Rule B)

- The Prior Prompt for this setup is: “The output of func should be derived from its first input parameter, ignoring the other.” This prior guides the model toward Rule A, forcing it to learn the calculation ($+2$) from the data distribution while enforcing the correct input variable ($x_1$) from the prior. We confirm that the ambiguous example quoted by the reviewer has been corrected in the text to clearly demonstrate the mechanism.

---

### Author Response · Authors · 2025-11-29
**Response to ICLR Reviewers-2: Enhancing Methodological Clarity and Generalizability of PGT-PGE**

### **2. Generalizability and Robustness of Negative Ablation and Priors (Reviewers x9pP, Tmjv, sFir)**

- **Ablation on the Negative Prior Loss (Reviewer Tmjv, x9pP)**: We conducted a controlled ablation study on the negative prior loss ($\mathcal{L}\_{-}$). The results demonstrate that the success of PGE strictly relies on the negative prior loss. We show that after removing the negative prior loss, the model degrades sharply compared to the standard PGE method. This confirms that the negative prior loss is essential for PGE to suppress spurious correlations and prune incorrect reasoning pathways (Table 4).

- **Robustness of Manually Written Priors (Reviewer x9pP, Tmjv)**: We also investigated manually generated priors and tested the robustness of different positive/negative prior prompts (Table 4). We found that our method maintains satisfactory performance across these settings. Additionally, PGE performs well on both manually written priors and those generated by SOTA LLMs, further verifying that PGE injects semantic knowledge into model parameters—enabling strong performance on diverse prior prompts that convey similar information and guidance. Notably, our PGT-PGE method is prior-based and unrelated to distillation methods, which primarily rely on data distributions distilled from SOTA models (similar to traditional methods in Figure 1). Furthermore, generating a single prior prompt incurs negligible computational cost (Table 4).

- **Priors Reappear during Inference Time (Reviewer sFir, x9pP)**: A key concern is whether reintroducing priors at inference time would improve performance. We conducted experiments comparing plain fine-tuning and PGT-PGE, and the marginal performance gain confirms that PGT-PGE has internalized prior knowledge into model parameters—resulting in consistent performance regardless of whether priors are present at inference. For synthetic benchmarks, learning requires both data distribution and priors: while plain fine-tuning only learns the ambiguous data distribution, PGT-PGE integrates prior-guided constraints. This explains why priors at inference provide little benefit to baseline models (Table 4).

### **3. Hyperparameter Selection ($\alpha, \beta, \gamma$) (Reviewers RksN, x9pP)**

We agree that the initial hyperparameter setting lacked sufficient analysis, and we have supplemented a detailed sensitivity study.

- **Sensitivity Analysis (Reviewer RksN, x9pP)**: Our new analysis investigates the impact of varying $\alpha, \beta, \gamma$ (weights for $\mathcal{L}, \mathcal{L}\_{+}, \mathcal{L}\_{-}$) on performance across tasks. We find that PGE is highly robust to variations in $\alpha$ (main task loss) and $\beta$ (positive prior loss) (Appendix A.5).

- **Recommended Procedure (Reviewer x9pP, RksN)**: We also present a practical hyperparameter selection strategy (Appendix A.7). Our findings suggest a simple, effective workflow for practitioners: set $\alpha=1$, $\gamma=0.1$ (standard weight), and $\beta=0.5$ as the initial value, then tune $\beta$ within a small range (e.g., $[0.3, 1.0]$) to adjust contrastive tension. This procedure significantly simplifies PGE's adoption for new tasks.

### **4. Methodological Clarity and Prompt Design (Reviewer RksN, sFir)**

- **Mechanism Clarification (Reviewer RksN)**: The core of PGE is the contrastive gradient term $\nabla_{\theta} (\mathcal{L}\_{+} - \mathcal{L}\_{-})$, which is added to the standard task gradient $\nabla\_{\theta} \mathcal{L}$. This encourages model parameters ($\theta$) to align with the positive prior ($\mathcal{L}\_{+}$) while actively diverging from the misleading negative prior ($\mathcal{L}\_{-}$). Priors are only used during training to compute these auxiliary losses and are completely removed at inference—confirming successful knowledge internalization.

- **Prompt Design Clarification (Figure 2) (Reviewer RksN, sFir)**: The updated Figure 2 provides a clearer visual illustration. "Prompt Design - Prior Injection" in the PGT paradigm refers to the structured concatenation of training input $x_i$ with positive prior $p_i^{+}$ and negative prior $p_i^{-}$, forming $x_{i}^{+}$ and $x_{i}^{-}$ respectively. This allows the model to compute the three distinct loss terms ($\mathcal{L}, \mathcal{L}_{+}, \mathcal{L}_{-}$) from a single training example.

---

### Author Response · Authors · 2025-11-29
**Response to ICLR Reviewers-3: Enhancing Methodological Clarity and Generalizability of PGT-PGE**

### **5. Response to Other Concerns**

- **Computational Cost of PGE (Reviewer Tmjv)**: We have further illustrated the computational cost of our method and explained why the approximately threefold increase in training computation is tolerable. The guidance from positive and negative priors helps the model converge quickly during training, so that fewer training epochs are required compared to baseline methods. In addition, computational costs during training contribute to the model's ability to generate without priors, which also avoids inference costs. (Appendix A.3)

- **The clarity of Figures and Tables (Reviewer sFir)**: We have updated a new illustration for our PGT paradigm in Figure 1 and revised the PGE method with a new example from the synthetic benchmark. We illustrate our PGT-PGE method on the synthetic benchmark (introduced in Section 2.2) instead of the original Binary Diagnosis of New Disease, where describing different diseases and corresponding priors explicitly is challenging. Moreover, we have refined our Results Table 2-4 according to the suggestions from the Review sFir to enhance clarity. Some model names and redundant columns have been removed to highlight important results.
We believe that these revisions and explanations directly address the primary concerns,substantially improving the paper’s clarity, generalizability, and evaluation scope. We are confident that the revised manuscript further reinforces the novelty and contributions of Prior-Guided Tuning.

We look forward to your further assessment.

Sincerely,

The Authors of Submission 15023

---

### Author Response · Authors · 2025-12-01
**Supplementary Rebuttal to Reviewers (ICLR Submission 15023)**

Dear Reviewers,

Following our previous revision and the submission of new experimental results—which addressed the primary concerns on clarity, robustness, and hyperparameter stability—we now focus on clarifying the scope and core design choices of our Prior-Guided Tuning (PGT) paradigm.

## **Response to Remaining Methodological and Scope Concerns**

### **1. Comparison to Dynamic Gradient Editing (Reviewer x9pP)**

We thank the reviewer for urging a clearer distinction from dynamic gradient methods. We emphasize that the core contribution of PGE is the PGT paradigm and the design of the contrastive prior loss, which steers the model towards specific semantic knowledge, distinct from balancing competing data-driven tasks.

- **Fixed-Weight Justification**: Unlike standard multi-task learning where tasks conflict arbitrarily, our three terms ($\mathcal{L}$, $\mathcal{L}\_{+}$, $\mathcal{L}\_{-}$) have a structured, non-conflicting relationship. This allows a simple, fixed-weight approach ($\alpha, \beta, \gamma$) to be effective and computationally economical, focusing capacity on internalizing the prior knowledge.

- **Future Extension**: Exploring dynamic weighting for even finer-grained control represents a valuable direction for future research.

### **2. Scope Extension to Generative Tasks (Reviewers Tmjv, sFir)**

We fully concur that demonstrating PGE’s ability to steer general generative tasks is a vital next step.

- **Core Claim Validation**: We clarify that our synthetic benchmark is already a simple generative task designed to resolve semantic ambiguity. Its success, alongside results on real-world tasks, fully validates the central hypothesis: that the PGT paradigm successfully steers back-propagation to internalize NL priors.

- **Distinction between Core Validity and Generalizability**: Extending this to large-scale, open-ended generative tasks primarily serves to demonstrate the generalizability of the mechanism, rather than impacting the validity of our core conclusion regarding the effectiveness of gradient-guided prior injection. This remains the highest-priority area for future work.

### **3. Comparison to Inference-Time Steering Methods (Reviewer Tmjv)**

- **Permanence vs. Transience: A Core Design Distinction**

Inference-time methods achieve steerability per-query, but the knowledge is never internalized into the weights ($\theta$). PGT-PGE is designed to permanently modify the model parameters to internalize the prior knowledge, eliminating the need for prior input or computational overhead at inference time, which is highly desirable for deployment.

### **4. PGE (Gradient Editing) vs. Contrastive Learning (Reviewer Tmjv)**

While PGE employs a contrastive-style loss, its mechanism fundamentally differs from standard Contrastive Learning (CL). CL aims to learn robust latent representations by regularizing the embedding space. In contrast, PGT-PGE uses the contrastive term applied to the gradients ($\nabla\_{\theta}$) to actively steer the task decision boundary in the weight space, thus injecting specific semantic knowledge and pruning incorrect reasoning pathways.

### **5. Controlled Ablation on Negative Prior Structure (Reviewer Tmjv)**

- **Utility vs. Practical Optimization**

We recognize the value of investigating the minimum viable semantic quality for the negative prior ($\mathcal{L}\_{-}$). Our existing ablations already confirm that the contrastive gradient term is essential and effective for our core claims. The reviewer’s suggestion is an excellent point for practical optimization (e.g., minimizing deployment cost), but does not affect the validity of our core methodological finding. This is a highly valuable area for our subsequent work.

### **6. Handling Multiple Natural Language Priors (Reviewer sFir)**

- **Extensibility of the PGT Framework**

This is an excellent question concerning real-world applicability. Our experiments focused on a single prior to isolate the mechanism, but the extension to multiple priors is a straightforward technical extension of our loss formulation.

- **Proposed Solution**

The PGE framework is inherently extensible to a set of multiple positive and negative priors via simple summation or averaging:

$\mathcal{L}\_{+} = \frac{1}{|\mathcal{P}\_{+}|} \sum\_{p^{+} \in \mathcal{P}\_{+}} \mathcal{L}(\mathbf{x}\_{i} \oplus p^{+}, y\_{i})$

$\mathcal{L}\_{-} = \frac{1}{|\mathcal{P}\_{-}|} \sum\_{p^{-} \in \mathcal{P}\_{-}} \mathcal{L}(\mathbf{x}\_{i} \oplus p^{-}, y\_{i})$

This confirms the practical feasibility of our framework.

We are confident that our comprehensive revisions and responses to these remaining methodological concerns further strengthen the novelty and contribution of Prior-Guided Tuning (PGT) and Prior-based Gradient Editing (PGE). We appreciate all suggestions, which we consider excellent priorities for future work.

We look forward to your final assessment.

Sincerely,

The Authors of Submission 15023

---

### Meta-Review · Area_Chair_cSs3 · 2025-12-30

**Summary:**

The paper presents Prior-based Gradient Editing (PGE), a method for incorporating natural-language priors into LLM fine-tuning via contrastive auxiliary losses. However, the reviews were predominantly negative: three of four reviewers recommended rejection (score: 2), while one gave a weak accept (score: 6). The primary concerns are as follows:

- **Unclear writing and definitions.** The submission lacks sufficient detail and clarity in several critical areas: the construction of the synthetic benchmark, the description of the core methodology, the design of the prompts, and the rationale for parameter selection.

- **Limited novelty.** The formulation of the optimization objective offers only incremental novelty for a top-tier venue.

- **Practical limitations of prior generation.** The efficacy of PGE is highly dependent on the quality of its priors, especially the negative prior. The reliance on powerful, proprietary LLMs (e.g., GPT-4o, DeepSeek-v3) to generate them introduces significant practical limitations, reducing the method's accessibility and generalizability.

- **Parameter sensitivity.** The key parameter β appears to be highly sensitive, as its optimal value varies across different experimental settings. This sensitivity is not sufficiently analyzed and raises concerns about the method's robustness.

The AC agrees that this work has potential but requires significant revision to address the major concerns raised by the reviewers for acceptance at a top-tier conference.

**Reviewer Concerns:**

While the authors have satisfactorily addressed most of the writing issues raised in the initial review, the core concerns regarding the method's practical limitations and parameter sensitivity remain unresolved.

**Reviewer Scores:**

From the AC's perspective, the three negative reviewers would not have changed their scores, even with full participation in the discussion.

---

### Decision · Program_Chairs · 2026-01-26

Reject